# Carrier-Mediated Process of Putrescine Elimination at the Rat Blood–Retinal Barrier

**DOI:** 10.3390/ijms24109003

**Published:** 2023-05-19

**Authors:** Yuma Tega, Yoshiyuki Kubo, Hiroaki Miura, Kairi Ri, Ayaka Tomise, Shin-ichi Akanuma, Ken-ichi Hosoya

**Affiliations:** 1Department of Pharmaceutics, Graduate School of Medicine and Pharmaceutical Sciences, University of Toyama, 2630 Sugitani, Toyama 930-0194, Japan; tega@pha.u-toyama.ac.jp (Y.T.);; 2Laboratory of Drug Disposition and Pharmacokinetics, Faculty of Pharma-Sciences, Teikyo University, Kaga 2-11-1, Tokyo 173-8605, Japan

**Keywords:** blood–retinal barrier, polyamine, putrescine, transport, choline transporter-like protein

## Abstract

Putrescine is a bioactive polyamine. Its retinal concentration is strictly controlled to maintain a healthy sense of vision. The present study investigated putrescine transport at the blood–retinal barrier (BRB) to gain a better understanding of the mechanisms of putrescine regulation in the retina. Our microdialysis study showed that the elimination rate constant during the terminal phase was significantly greater (1.90-fold) than that of [^14^C]D-mannitol, which is a bulk flow marker. The difference in the apparent elimination rate constants of [^3^H]putrescine and [^14^C]D-mannitol was significantly decreased by unlabeled putrescine and spermine, suggesting active putrescine transport from the retina to the blood across the BRB. Our study using model cell lines of the inner and outer BRB showed that [^3^H]putrescine transport was time-, temperature-, and concentration-dependent, suggesting the involvement of carrier-mediated processes in putrescine transport at the inner and outer BRB. [^3^H]Putrescine transport was significantly reduced under Na^+^-free, Cl^−^-free, and K^+^-replacement conditions, and attenuated by polyamines or organic cations such as choline, a choline transporter-like protein (CTL) substrate. Rat CTL1 cRNA-injected oocytes exhibited marked alterations in [^3^H]putrescine uptake, and CTL1 knockdown significantly reduced [^3^H]putrescine uptake in model cell lines, suggesting the possible participation of CTL1 in putrescine transport at the BRB.

## 1. Introduction

Polyamines are essential compounds that play an important role in biological processes, including cell proliferation and differentiation [1,2]. Disruption of polyamine homeostasis has been considered to affect health and lifespan. Indeed, previous research has suggested that reductions in polyamine are related to age-associated pathology and mortality, which are improved by polyamine treatment, affecting DNA methylation [3]. In addition, severe diseases, such as cancer, multiple sclerosis, psoriasis, chronic renal failure, and Duchenne muscular dystrophy, have been reported to cause a significant change in polyamine concentration in cells, blood, or urine [4,5,6,7,8]. In a study on cancer, the modulation of polyamine concentration was suggested to affect the state and progression of the disease, since the inhibitor of the polyamine biosynthetic pathway suppressed the proliferation of cancer cells [9,10]. These cumulative findings suggest that dysregulated polyamine concentrations may be closely associated with the pathogenesis of diseases.

The physiological and pathological roles of polyamines have been extensively investigated in various tissues, including the retina, which is essential for the sense of vision. A deficiency in polyamines was reported to cause the loss of photoreceptor cells in the developing rabbit eye [11], suggesting their importance in retinal cell development. On the other hand, a study on bovine eyes reported the induction of apoptosis by a polyamine, spermine, in retinal pigment epithelial cells [12]. Several reports have indicated an association of polyamines with retinal degeneration in multiple sclerosis and gyrate atrophy of the choroid and retina [12,13]. These lines of evidence clearly suggested the importance of regulating polyamine concentrations in the retina.

Putrescine, which is synthesized from L-ornithine by ornithine decarboxylase, is the first polyamine product of the polyamine biosynthetic pathway. It is reported that putrescine is converted into γ-aminobutyric acid (GABA), and plays a role in synapse formation and neurodevelopment in the retinas of postnatal rats [14]. On the other hand, excessive putrescine facilitates the entry of Ca^2+^ through N-methyl-D-aspartate-type glutamate receptors and enhances excitotoxicity in retinal ganglion cells [15]. Furthermore, putrescine can be enzymatically converted to spermine [16], which is associated with apoptosis and degeneration in retinal cells [12,15]. Hence, the concentration of putrescine must be strictly controlled in the retina.

Putrescine and other polyamines are rarely transported via passive diffusion across the lipid bilayer, since they exist in cationic forms at a physiological pH (pH 7.4). Hence, specific transport systems are needed for the translocation of polyamines between intracellular and extracellular spaces. Indeed, the carrier-mediated transport of putrescine has been reported in bovine lymphocytes and cultured mouse astrocytes [17,18]. The retina is separated from the blood by retinal capillary endothelial and retinal pigment epithelial cells, composing the inner and outer blood–retinal barriers (BRBs), respectively. These regulate the permeation of compounds such as drugs, nutrients, and metabolites between the retina and the blood by forming tight junctions and expressing various transporters [19]. Previously, our study on spermine, which is the final product of the polyamine biosynthetic pathway, suggested the presence of an efflux transport system from the retina and cerebrospinal fluid across the BRB and the blood–cerebrospinal fluid barrier, respectively [20,21]. In addition, previous in vitro studies using a model cell of the BRB showed that the carrier-mediated transport of spermine interacted with other polyamines [21], implying the existence of a clearance system for polyamines at the BRB. However, for polyamines other than spermine, the mechanisms of elimination have not yet been fully elucidated. It is possible that the efflux transport system of putrescine at the BRB also plays a role in the modulation of retinal distribution to prevent the adverse effects of polyamines in the retina.

In the present study, in vivo and in vitro studies were performed to examine the elimination of putrescine from the retina to the circulating blood. In the in vivo study, the transport of putrescine from the retina to the blood was analyzed by means of microdialysis. In the in vitro study, the transport of putrescine at the inner and outer BRB was analyzed in their in vitro model cell lines, TR-iBRB2 cells and RPE-J cells, respectively [22,23].

## 2. Results

### 2.1. In Vivo Studies of [^3^H]putrescine Transport in Rats

After an intravitreal bolus injection, microdialysis of the [^3^H]putrescine remaining in the vitreous humor was carried out in rats, and its time profile showed a biexponential pattern of elimination from the vitreous humor (Figure 1A), exhibiting an apparent elimination constant (β) during the terminal phase of 19.4 × 10^−3^ ± 2.8 × 10^−3^ min^−1^ (Figure 1B). In rats, [^14^C]D-mannitol was also examined as a bulk flow marker of passage from the vitreous humor to Schlemm’s canal and/or the uveoscleral outflow route [21,24], and its biexponential elimination exhibited a β value of 10.2 × 10^−3^ ± 1.8 × 10^−3^ min^−1^ during the terminal phase (Figure 1B). Moreover, [^3^H]putrescine showed a significantly higher β value (1.90-fold) compared with [^14^C]D-mannitol, indicating that [^3^H]putrescine was eliminated from the vitreous humor faster than [^14^C]D-mannitol. In addition, the difference in the β value between [^3^H]putrescine and [^14^C]D-mannitol (Δβ) decreased significantly, by 70.5% and 67.6%, in the presence of unlabeled putrescine and spermine, respectively. On the other hand, no significant effect was shown for tetraethylammonium (TEA) (Table 1).

The carotid artery single-injection method showed that [^3^H]putrescine had an retinal uptake index (RUI) value of 41.6%; this was reduced to 33.5%, which was 80.5% of the control, in the presence of 50 mM unlabeled putrescine (Table 2).

### 2.2. In Vitro Studies of [^3^H]putrescine Transport in TR-iBRB2 Cells

The transport of [^3^H]putrescine in TR-iBRB2 cells was examined at 37 °C. A time-dependent increase in [^3^H]putrescine uptake was observed for at least 15 min, with an initial uptake rate of 6.65 ± 0.48 μL/(min·mg protein), and a significant reduction of 92.9% in the transport of [^3^H]putrescine was shown at 5 min under 4 °C (Figure 2A). In addition, TR-iBRB2 cells showed concentration-dependent [^3^H]putrescine transport, and a K_m_ of 139 ± 18 μM and a V_max_ of 482 ± 40 pmol/(min·mg protein) were estimated (Figure 2B). The transport of [^3^H]putrescine by TR-iBRB2 cells was reduced significantly by 70.8% and 34.4% at pH 6.4 and pH 8.4, respectively (Figure 2C), and the study of different ionic conditions exhibited significant reductions of 38.7%, 32.6%, and 78.3% under Na^+^-free, Cl^−^-free, and K^+^-replacement conditions, respectively (Figure 2D).

In the in vitro analysis of inhibition, the transport of [^3^H]putrescine by TR-iBRB2 cells was markedly reduced in the presence of polyamines, including putrescine, spermidine, spermine, and agmatine. When the effects of the typical substrates and inhibitors of organic cation transporters were investigated, the transport of [^3^H]putrescine in TR-iBRB2 cells decreased significantly in the presence of serotonin, TEA, L-carnitine, choline, L-ornithine, 1-methyl-4-phenylpyridinium (MPP^+^), L-arginine, and cimetidine (Table 3).

### 2.3. In Vitro Studies of [^3^H]putrescine Transport in RPE-J Cells

The transport of [^3^H]putrescine in RPE-J cells was examined. The transport showed a time-dependent increase for at least 10 min, and a significant reduction of 70.6% was shown at 10 min at 4 °C (Figure 3A). RPE-J cells showed concentration-dependent transport of [^3^H]putrescine with a K_m_ of 3.78 ± 0.67 μM, a V_max_ of 29.3 ± 4.2 pmol/(min·mg protein), and a K_d_ of 1.57 ± 0.09 μL/(min·mg protein) (Figure 3B). The transport of [^3^H]putrescine by RPE-J cells increased significantly by 38.1% at pH 8.4, while no significant effect was observed at pH 6.4 (Figure 3C). In addition, the transport of [^3^H]putrescine in RPE-J cells decreased significantly by 23.1%, 29.8%, and 39.3% under Na^+^-free, Cl^−^-free, and K^+^-replacement conditions, respectively (Figure 3D).

The transport of [^3^H]putrescine by RPE-J cells was markedly reduced in the presence of polyamines, serotonin, choline, and pyrimethamine, and also decreased significantly in the presence of TEA, L-carnitine, MPP^+^, cimetidine, lopinavir, and fluvoxamine; no inhibitory effect was observed in the presence of L-ornithine and L-arginine (Table 3).

In the study of directional [^3^H]putrescine uptake by RPE-J cells cultured on collagen-coated Transwell^®^ inserts (Corning, Corning, NY, USA), the A-to-C and B-to-C uptake values of [^3^H]putrescine at 30 min were 16.8 μL/mg protein and 18.9 μL/mg protein, respectively, and they decreased significantly in the presence of unlabeled putrescine (Figure 4).

### 2.4. Study of Uptake in Xenopus laevis Oocytes

We previously reported that TR-iBRB2 cells express choline transporter-like proteins (CTLs), namely CTL1, -3, and -4, at the mRNA level. In particular, 213 times more CTL1 mRNA was expressed than CTL3 and CTL4 mRNA in TR-iBRB2 cells [25]. However, the role of CTL1 at the inner BRB is still unclear, since the knockdown study showed a minor role of CTL1 in choline transport in TR-iBRB2 cells [25]. The uptake of choline in *Xenopus laevis* oocytes injected with *Torpedo marmorata* CTL1 cRNA and COS7 cells expressing mouse CTL1 was reported to be weakly or moderately Na^+^-dependent [26,27], similar to the present results on the uptake of [^3^H]putrescine by TR-iBRB2 and RPE-J cells. In addition, the uptake of [^3^H]putrescine by TR-iBRB2 and RPE-J cells was inhibited by choline. These results imply the possible involvement of CTL in putrescine transport at the BRB. Hence, the putrescine transport abilities of rCTL1, rCTL3, and rCTL4 were examined by means of the oocyte expression system. The oocytes injected with rCTL1 cRNA exhibited a remarkable decrease of 87.9% in [^3^H]putrescine uptake compared with water-injected oocytes (Figure 5), while no significant alteration was observed in [^3^H]putrescine uptake by oocytes injected with rCTL3 cRNA and rCTL4 cRNA (Figure 5). In addition, oocytes injected with rCTL1 cRNA, rCTL3 cRNA, and rCTL4 cRNA exhibited a significant decrease of 63.1%, 43.6%, and 43.2%, respectively, in the uptake of [^3^H]spermine compared with water-injected oocytes (Figure 5). The oocytes injected with rCTL cRNA exhibited nonsignificant or only slight alterations in other cationic compounds such as [^3^H]verapamil, [^3^H]nicotine, [^3^H]pyrilamine, and [^3^H]clonidine (Appendix A).

### 2.5. Knockdown Analysis of rCTL1

To clarify the involvement of rCTL1 in putrescine transport, a knockdown analysis was performed. In TR-iBRB2 cells, quantitative real-time polymerase chain reaction (PCR) analysis showed that the transfection of siRNA designated for rCTL1 significantly reduced the mRNA expression levels of rCTL1 by 70.2% (Figure 6A). In the uptake analysis, the siRNA designated for rCTL1 significantly reduced [^3^H]putrescine uptake by 70.5%(Figure 6B), while it showed a smaller reduction of 19.8% in [^3^H]taurine uptake (Appendix A).

The siRNA designated for rCTL1 significantly reduced the mRNA expression levels of rCTL1 and [^3^H]putrescine uptake by 44.0% and 56.5%, respectively, in RPE-J cells (Figure 6A,B), while no significant effect was shown in [^3^H]taurine uptake (Appendix A).

## 3. Discussion

Putrescine is a polyamine that is distributed ubiquitously in the body, and it is associated with retinal function and pathogenesis, such as GABA synthesis in the developing retina and the enhancement of excitotoxicity in the retinal ganglion cells [14,15]. In addition, putrescine has been reported to block ionotropic glutamate receptors and to play an important role in excitatory neurotransmission [28]. These reports indicate the importance of precisely regulating the concentration of putrescine to maintain a healthy retina. In particular, to avoid the adverse effects of putrescine, a better understanding of the mechanism of eliminating putrescine from the retina is needed. Hence, in the present study, the transport of putrescine from the retina to the blood at the inner and outer BRB was investigated by means of in vivo and in vitro approaches. We found that putrescine is eliminated via a carrier-mediated transport system from the retina, and that rCTL1 may be involved in putrescine transport at both the inner and outer BRB.

A microdialysis study was carried out to investigate the retina-to-blood transport of [^3^H]putrescine at the BRB, and [^14^C]D-mannitol was adopted as a bulk flow marker. As a matter of consideration for the microdialysis study, it was possible that the putrescine was metabolized and eliminated from the retina as a metabolite, since putrescine can be converted to spermidine and spermine in the retina [16], and spermine undergoes efflux from the retina via a carrier-mediated transport process [21]. To investigate this possibility, we analyzed the [^3^H]putrescine in the retina using the high performance liquid chromatography method after the microdialysis study, and found that most of the radioactivity in the retina was derived from [^3^H]putrescine (Appendix A). This result suggests that the [^3^H]putrescine injected into the vitreous humor remained intact during the microdialysis study. The time profile of the concentration of [^3^H]putrescine in the dialysate showed a biexponential reduction in the vitreous humor, with the initial slope (α) being steeper than the later slope (β); these initial and later declines are thought to represent diffusion into the vitreous humor after the vitreous bolus injection and the elimination of [^3^H]putrescine from the vitreous humor, respectively (Figure 1A). The β value of [^3^H]putrescine (19.4 × 10^−3^ min^−1^) was greater than that of [^14^C]D-mannitol (10.2 × 10^−3^ min^−1^), which suggests that [^3^H]putrescine underwent retina-to-blood efflux transport across the BRB, as well as elimination from the vitreous humor via passive diffusion and bulk flow (Figure 1B). In addition, the microdialysis study suggested that polyamine-sensitive and carrier-mediated transport was involved in the retina-to-blood transport of [^3^H]putrescine at the BRB, because the Δβ value decreased significantly in the presence of unlabeled spermine and putrescine, although no alteration was shown in the presence of TEA (Table 1). On the other hand, the study with a single carotid injection of [^3^H]putrescine showed only 20% inhibition of the retinal uptake of [^3^H]putrescine by unlabeled putrescine (Table 2). Although the carrier-mediated transport process is suggested to partially participate in the influx transport of putrescine to the retina, the result emphasized the involvement of carrier-mediated transport in the efflux of [^3^H]putrescine rather than influx at the BRB.

The in vivo study demonstrated that the elimination of putrescine from the retina involved a retina-to-blood carrier-mediated transported process. However, in vivo studies cannot assess the transport of putrescine at the inner and outer BRB separately. To investigate the detailed mechanism of putrescine transport at the inner and outer BRB, a transport study was carried out with in vitro model cell lines, namely TR-iBRB2 cells and RPE-J cells, respectively. The study of the TR-iBRB2 cells suggested the involvement of a carrier-mediated process of putrescine transport at the inner BRB, since [^3^H]putrescine transport took place in a time-, temperature- and concentration-dependent manner, with a K_m_ of 139 μM (Figure 2). In addition, the study suggested that the carrier-mediated putrescine transport process at the inner BRB was sensitive to pH, Na^+^, Cl^−^, and membrane potential, since the transport of [^3^H]putrescine by TR-iBRB2 cells decreased significantly under different pH conditions, and under Na^+^-free, Cl^−^-free, and K^+^-replacement conditions (Figure 2).

Similarly, a study of RPE-J cells suggested the involvement of carrier-mediated putrescine transport at the outer BRB because the RPE-J cells exhibited dependence on the time, temperature, and concentration, with a K_m_ of 3.78 μM (Figure 3). In addition, the study suggested that the carrier-mediated process of putrescine transport at the outer BRB was sensitive to Na^+^, Cl^−^, and membrane potential, since the transport of [^3^H]putrescine by RPE-J cells decreased significantly under Na^+^-free, Cl^−^-free, and K^+^-replacement conditions (Figure 3). RPE-J cells are known to be polarized cells, and a study using collagen-coated Transwell^®^ inserts was used to assess the uptake of compounds from both the blood and retina sides. The uptake of [^3^H]putrescine by the RPE-J cells on the Transwell^®^ inserts suggested the carrier-mediated transport of putrescine in the blood- and retina-side plasma membranes of RPE-J cells, since the comparable A-to-C and B-to-C uptake of [^3^H]putrescine was inhibited in the presence of unlabeled putrescine (Figure 4). This result suggests that the carrier-mediated transport process at the outer BRB is at least partially responsible for both the influx and efflux transport of putrescine.

The inhibition study implied the involvement of organic cation transporters, such as OCTs, OCTNs, PMAT, and MATE1, in the retina-to-blood transport of putrescine at the inner and outer BRB, since the transport of [^3^H]putrescine by TR-iBRB2 cells and RPE-J cells was significantly inhibited by cationic compounds, such as serotonin, TEA, L-carnitine, choline, and MPP^+^; moreover, this study marked inhibition in the presence of polyamines, including putrescine, spermidine, spermine, and agmatine (Table 3). However, previous works have reported that the expression of OCT mRNA is not detectable in the inner BRB [25], and that putrescine is not recognized as a substrate by OCT1 and OCT3 [29]. OCT2 has been reported to have a low affinity for putrescine, with a K_m_ of 7.5 mM [30], which clearly differs from that estimated in TR-iBRB2 cells (139 μM) and RPE-J cells (3.78 μM). Regarding OCTN2 and MATE1, their prominent interaction with cimetidine has been reported [31,32], while in this study, nonsignificant and moderate effects of cimetidine were shown in TR-iBRB2 cells and RPE-J cells, respectively. In addition, putrescine and agmatine have been reported to have no significant effects on the substrate transport of OCTN1 and PMAT, respectively [33,34], while in this study, these compounds clearly inhibited the transport of [^3^H]putrescine by TR-iBRB2 cells and RPE-J cells. According to these results, organic cation transporters other than OCTs, OCTNs, MATE, and PMAT mediate the uptake of [^3^H]putrescine by TR-iBRB2 cells and RPE-J cells.

The in vitro study suggested a minor contribution of well-known organic cation transporters to the transport of putrescine at the BRB. We focused on CTLs as a candidate for the putrescine transporter, since choline inhibited the uptake of [^3^H]putrescine by both TR-iBRB2 cells and RPE-J cells (Table 3). In addition, we reported the mRNA expression of CTL1, -3, and -4 in TR-iBRB2 cells [25] and confirmed that rCTL1, but not rCTL2-4, was expressed at the mRNA level in RPE-J cells (Appendix A). In the uptake study with *Xenopus laevis* oocytes, the details of the mechanism by which the injection of cRNA reduced polyamine uptake were unclear; however, in this paper, the significantly changed uptake of [^3^H]putrescine and [^3^H]spermine was observed in rCTL-expressing oocytes (Figure 5), suggesting the possible involvement of rCTLs in polyamine transport. In particular, rCTL1-expressing oocytes showed a marked reduction in [^3^H]putrescine uptake, but no significant change in the uptake of [^3^H]putrescine by rCTL3- or rCTL4-expressing oocytes was seen. In addition, the present study showed that rCTL1 knockdown significantly decreased the mRNA expression levels of rCTL1 and the transport of [^3^H]putrescine in TR-iBRB2 cells and RPE-J cells, without an alteration in their [^3^H]taurine transport (Figure 6 and Appendix A). These results suggest that rCTL1 may play a role in the transport of putrescine at the inner and outer BRB.

Previous studies reported that the putrescine concentrations in the retina and plasma were 89 μM and 1.03 μM, respectively, and the retina-to-plasma concentration ratio was calculated to be 86 [16,35], suggesting that the supply of putrescine from the blood and/or its synthesis are predominant compared with the elimination of putrescine from the retina via metabolism and efflux transport. However, the present study showed the elimination of intact [^3^H]putrescine from the retina (Appendix A), implying that the efflux transport process of putrescine plays at least a partial a role in the regulation of in vivo putrescine concentrations in the retina. In vitro gain and loss of function studies have suggested that rCTL1 may play an important role in the transport of putrescine in retinal capillary endothelial cells and retinal pigment epithelial cells. CTL1 is possibly involved in the efflux transport of putrescine at the inner and outer BRB. However, the localization of CTL1 at the BRB has not been clarified yet, and future studies are needed to clarify the direct involvement of CTL1 in the efflux transport of putrescine at the inner and outer BRB.

In conclusion, we suggest that putrescine is eliminated from the retina to the blood across the inner and outer BRB through a carrier-mediated transport process, and that CTL1 transports putrescine. In vivo and in vitro studies showed the possible involvement of CTL1 in the efflux transport of putrescine at the inner and outer BRB. The present findings will help us to develop an understanding of the mechanism of regulating retinal polyamine concentrations and the pathogenesis of retinal diseases caused by disrupted polyamine homeostasis.

## 4. Materials and Methods

### 4.1. Reagents, Animals, and Cells

Chemicals of reagent grade, which were used in the present study, were commercially available. Specifically, [2,3-^3^H(N)]putrescine ([^3^H]putrescine), *n*-[1-^14^C]butanol ([^14^C]*n*-butanol), [1-^14^C]D-mannitol, [N-methyl-^3^H]verapamil ([^3^H]verapamil), and [terminal methylene-^3^H]spermine ([^3^H]spermine) were obtained from American Radiolabeled Chemicals (St. Luis, MO, USA), whereas [pyridinyl-5-^3^H]pyrilamine ([^3^H]pyrilamine), [benzene ring-^3^H]clonidine hydrochloride ([^3^H]clonidine), L-(-)-[N-methyl-3H] nicotine ([3H]nicotine) and [3H]taurine were purchased from PerkinElmer (Waltham, MA, USA). The polyethylene tubing and an infusion pump were obtained from Natsume (Tokyo, Japan) and Harvard (Holliston, MA, USA), respectively, and the TEP-50 microdialysis probe was purchased from Eicom (Kyoto, Japan).

Male Wistar/ST rats were purchased from Japan SLC (Hamamatsu, Japan), and were used according to the standards of the Association for Research in Vision and Ophthalmology (ARVO) and the guidelines of the University of Toyama for animal experiments. The approved numbers for the animal studies are shown in the Institutional Review Board Statement section.

In the present in vitro study, TR-iBRB2 cells and RPE-J cells were adopted as the in vitro model cell lines of the inner and outer BRB, respectively [22,23]. Dulbecco’s modified Eagle’s medium (DMEM) and BioCoat Collagen I Cellware 24-well dishes were purchased from Nissui Pharmaceuticals (Tokyo, Japan) and BD Bioscience (Franklin Lakes, NJ, USA), respectively. The TR-iBRB2 cells and RPE-J cells were cultured in DMEM, as described elsewhere [22,23]. The DC protein assay kit was purchased from Bio-Rad (Hercules, CA, USA), and was used for measuring the cellular protein content. Furthermore, details of the composition of the culture medium and buffer solutions, such as the extracellular fluid (ECF) buffer, are shown in the Appendix A.

### 4.2. Microdialysis of [^3^H]putrescine in Rats

As described in previous reports [21,24], microdialysis of [^3^H]putrescine was performed. In brief, an eight-week-old male Wistar/ST rat (250–300 g) was anesthetized with sodium pentobarbital (50 mg/kg) and placed on a stereotaxic frame (Narishige, Tokyo, Japan). Local anesthetization of the rat’s eyelids was performed by instilling 2% xylocaine to prevent eye blinking, and a Ringer–HEPES solution (1 μL, pH 7.4) including [^3^H]putrescine (2.0 μCi) and [^14^C]D-mannitol (0.2 μCi) was injected with a microsyringe (Hamilton Company, Reno, NV, USA) at a depth of 3.0 mm from the ocular surface, followed by the implantation of a microdialysis probe (TEP-50) in the vitreous chamber. After fixation of the microdialysis probe on the ocular surface with surgical glue (Daiichi Sankyo, Tokyo, Japan), the Ringer–HEPES solution was continuously supplied to the probe (2 μL/min, 37 °C) by means of polyethylene tubing (Natsume, Osaka, Japan) and an infusion pump (Harvard). A liquid scintillation counter (AccuFLEX LSC-7400, Nippon RayTech Co., Ltd., Tokyo, Japan) was used to determine the radioactivity in the dialysate, which was collected at designated times. When the effects of the compounds were examined, the Ringer–HEPES solution containing 50 mM of the compound was substituted for the Ringer–HEPES solution.

To analyze the data, according to previous reports [21,24], Equation (1) was used to calculate the vitreous concentrations normalized by the injected dose (C_P_, % of dose/mL) from the total radioactivity in the injectate (Dose_tracer_, dpm) and the concentration in the dialysate (C_T_, dpm/mL). The C_P_ at time t was defined as C_P_(t), and the nonlinear least-squares regression analysis program MULTI [36] was adopted to fit C_P_ to Equation (2), which includes the apparent first-order rate constants for the initial phase (α) and the terminal phase (β), and the intercepts on the *y*-axis for each exponential segment (A and B).
C_P_ = C_T_/Dose_tracer_ × 100(1)
C_P_(t) = A × e^−αt^ + B × e^−βt^(2)
Recovery (%) = C_T_/C_V_ × 100(3)
(Percentage of control) = (Δβ in the presence of inhibitor)/(Δβ in the absence of inhibitor) × 100(4)

In Equation (3), the concentrations in the test solution (C_V_, dpm/mL) and the dialysate (C_T_, dpm/mL) were used to estimate the probe’s recovery, and the recoveries of [^3^H]putrescine (9.42%) and [^14^C]D-mannitol (7.01%) were constant over 180 min in the present study.

In the present study, the difference in the β values between [^3^H]putrescine and [^14^C]D-mannitol was defined as Δβ. Equation (4) was used to express the inhibitory effect on the elimination of [^3^H]putrescine from the vitreous humor as a percentage of the control.

### 4.3. Retinal Uptake Index (RUI) of [^3^H]putrescine in Rats

The RUI of [^3^H]putrescine was investigated in male Wistar/ST rats (six weeks old) using [^14^C]*n*-butanol as a freely diffusible internal reference, as described elsewhere [21]. In brief, with Ringer–HEPES buffer as the vehicle, [^3^H]putrescine (5 μCi/rat) and [^14^C]*n*-butanol (0.5 μCi/rat) were injected into the common carotid arteries of rats that had been anesthetized with pentobarbital. Fifteen seconds after injection, the retinal tissue was collected after decapitation of the rats, followed by lysis with 2 N sodium hydroxide and neutralization with 2 N hydrochloric acid. The sample’s radioactivity was determined by means of AccuFLEX LSC-7400 (Nippon RayTech Co., Ltd.), and the RUI was calculated using Equation (5).
RUI = 100 × ([^3^H]/[^14^C] (dpm in the tissue))/([^3^H]/[^14^C] (dpm in the injectate))(5)

### 4.4. Analysis of the Uptake of [^3^H]putrescine by Cells

Before the uptake assay, TR-iBRB2 cells and RPE-J cells cultured on a BioCoat Collagen I Cellware 24-well dish (BD Bioscience, Franklin Lakes, NJ, USA) were rinsed with an ECF buffer three times. The uptake assay was initiated by adding 200 μL of the ECF buffer containing [^3^H]putrescine (0.1 μCi) at 37 °C, as described elsewhere [21,24]. After the termination of the assay, the cells were lysed with 1 N sodium hydroxide and neutralized with 1 N hydrochloric acid, and the radioactivity was determined using a liquid scintillation counter (AccuFLEX LSC-7400). As described previously [21,24], the cellular uptake of [^3^H]putrescine was expressed by means of the cell-to-medium (cell/medium) ratio, which was determined using Equation (6), and MULTI was used to fit the data obtained from the TR-iBRB2 cells and RPE-J cells via Equations (7) and (8), respectively. In these two equations, S, K_m_, V, V_max_, and K_d_ represent the concentration of putrescine, the Michaelis constant, the uptake rate, the maximal uptake rate, and the nonsaturable uptake rate constant, respectively.
Cell/medium ratio = ([^3^H] dpm per cell protein (mg))/([^3^H] dpm per μL medium)(6)
V = V_max_ × S/(K_m_ + S)(7)
V = V_max_ × S/(K_m_ + S) + K_d_ × S(8)

When the directional transport of [^3^H]putrescine by RPE-J cells was examined, RPE-J cells were cultured for 6 days at 33 °C with the normal medium, and then, for another 2 days with 10 nM of a medium containing all trans-retinoic acids, on Transwell^®^ polyester membrane inserts (pore size: 0.4 μm, 12 mm diameter, Corning).

### 4.5. Uptake Study with Xenopus laevis Oocytes

The uptake of radiolabeled compounds, such as [^3^H]putrescine and [^3^H]spermine, by cRNA-injected oocytes was carried out as described elsewhere [37,38]. In brief, the RiboMAX Large-Scale RNA Production System-T7 (Promega, Madison, WI, USA) was adopted to synthesize capped cRNA using a pGEM-HE vector harboring the transporter cDNA. With RNase-free water as a vehicle, the capped cRNA (10 ng) was injected into the oocytes. After three days of incubation in a standard oocyte saline (SOS) solution with antibiotics at 18 °C, an uptake study was carried out in an ND96 solution at 20 °C. The assay was initiated by adding 200 µL of ND96 containing 0.5 µCi [^3^H]putrescine, and was terminated by adding an ice-cold ND96 solution at the designated time. The oocytes were rinsed with ice-cold ND96, followed by lysis in 5% sodium dodecyl sulfate, and their radioactivity was determined using an AccuFLEX LSC-7400 (Nippon RayTech Co., Ltd.). According to previous reports [37,38], the uptake of [^3^H]putrescine by oocytes was expressed as the oocyte-to-medium (oocyte/medium) ratio, which was calculated using Equation (9).
Oocyte/medium ratio = ([^3^H] dpm per oocyte)/([^3^H] dpm per μL medium)(9)

In the present study, the full-length cDNA of rCTL3 and rCTL4 was cloned from the total RNA extracted from TR-iBRB2 cells, and the full-length cDNA of rCTL1 was also used as reported previously [25]. Full-length cDNA cloning was carried out with ReverTraAce (TOYOBO, Osaka, Japan), oligo dT primer, and KOD FX Neo (TOYOBO); the primers are shown in Appendix A. Their nucleic acid sequences were confirmed by means of an ABI PRISM 3130 Genetic Analyzer (Thermo Fisher Scientific, Waltham, MA, USA).

### 4.6. Knockdown Study

As described in our previous work [25], a knockdown study was carried out using small interfering RNA (siRNA) designed for rCTL1 and Stealth RNAi Negative Control Medium GC Duplexes, which was purchased from Thermo Fisher Scientific.

The transfection of siRNA into cells cultured on a 6-well plate was carried out in accordance with the manufacturer’s instructions, and siRNA (90 pmol/well) was introduced to the cells using Lipofectamine™ RNAiMAX (9 μL/well, Thermo Fisher Scientific). Uptake studies of [^3^H]putrescine were performed 48 h after the transfection.

The efficiency of the knockdown was determined via an analysis of mRNA expression using an Mx-3000P (Agilent Technologies, Santa Clara, CA, USA), a quantitative real-time PCR system. The nucleic acid alignments of the primers for the real-time PCR were as follows: for rCTL1 (Slc44a1, NM_053492), the forward primer was 5′-CATGTGGTGGTACCACGTGGTC-3′ and the reverse primer was 5′-CGAATAAGGGGGTTCACTGC-3′, whereas for β-actin (NM_031144), the forward primer was 5′-TCATGAAGTGTGTGACGTTGACATCCGT-3′ and the reverse primer was 5′-CCTAGAAGCATTTGCGGTGCACGATG-3′. SYBR^®^ Premix Ex Taq™ (Takara, Shiga, Japan) was used for the real-time PCR, for which the conditions were as follows: 95 °C for 30 s, 65 °C for 30 s, and 72 °C for 30 s, with 40 cycles. The initial amount of rCTL1 transcript was extrapolated using a determined number of threshold cycles. A standard curve was produced using control plasmids carrying the gene of interest.

The primers for RT-PCR analysis of CTLs with RPE-J cells are shown in Appendix A.

## Figures and Tables

**Figure 1 ijms-24-09003-f001:**
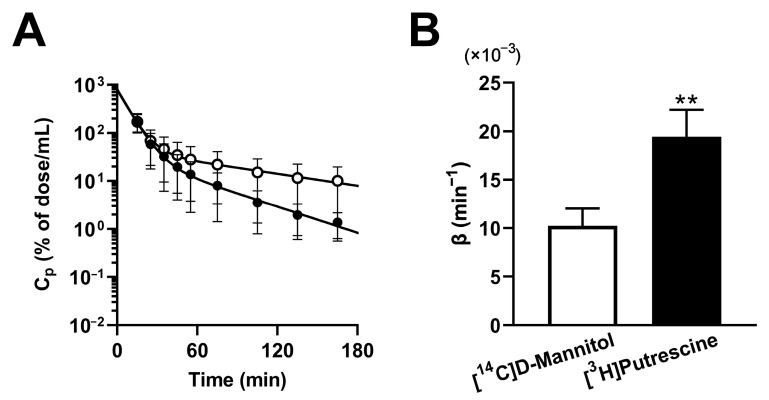
In vivo microdialysis of [^3^H]putrescine. Time profile of [^3^H]putrescine and [^14^C]D-mannitol in the vitreous humor after a vitreous bolus injection (**A**). Closed and open circles represent the concentration in the dialysate of [^3^H]putrescine and [^14^C]D-mannitol, respectively. Elimination rate constants (β) of [^3^H]putrescine and [^14^C]D-mannitol in the terminal phase (**B**). The points and columns represent the means ± S.D. (*n* = 4). ** *p* < 0.01 vs. [^14^C]D-mannitol.

**Figure 2 ijms-24-09003-f002:**
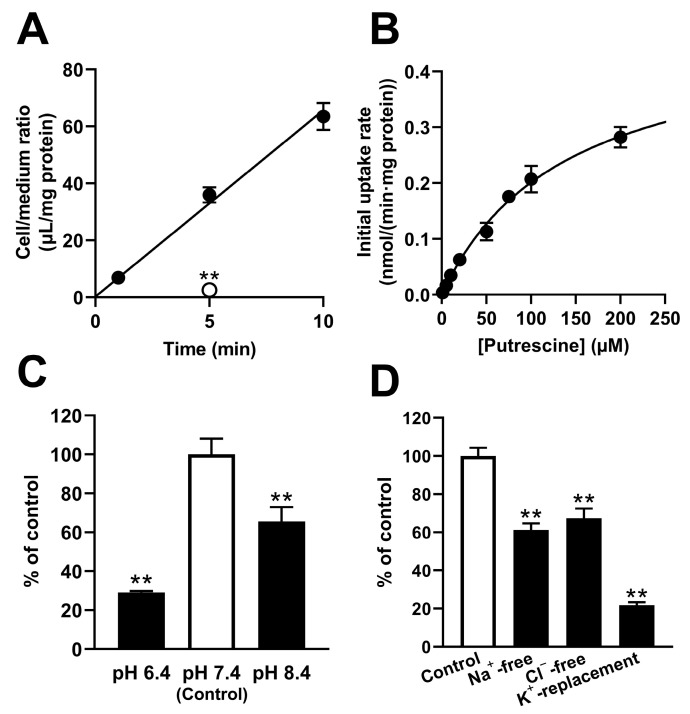
Uptake of [^3^H]putrescine by TR-iBRB2 cells. (**A**) Uptake of [^3^H]putrescine (0.1 μCi) was examined at 37 °C (closed circles), and its dependence on temperature was examined at 4 °C (open circles). (**B**) The concentration-dependent uptake of [^3^H]putrescine (0.1 μCi) was assessed at 37 °C for 5 min at concentrations in the range of 1 μM to 200 μM. The data obtained in the uptake assay were plotted in the Michaelis–Menten analysis. The dependence of [^3^H]putrescine uptake on extracellular pH (**C**), and extracellular Na^+^, Cl^−^, and membrane potential (**D**), was examined at 37 °C for 5 min. The points and columns represent the means ± S.D. (*n* = 3). ** *p* < 0.01 vs. control.

**Figure 3 ijms-24-09003-f003:**
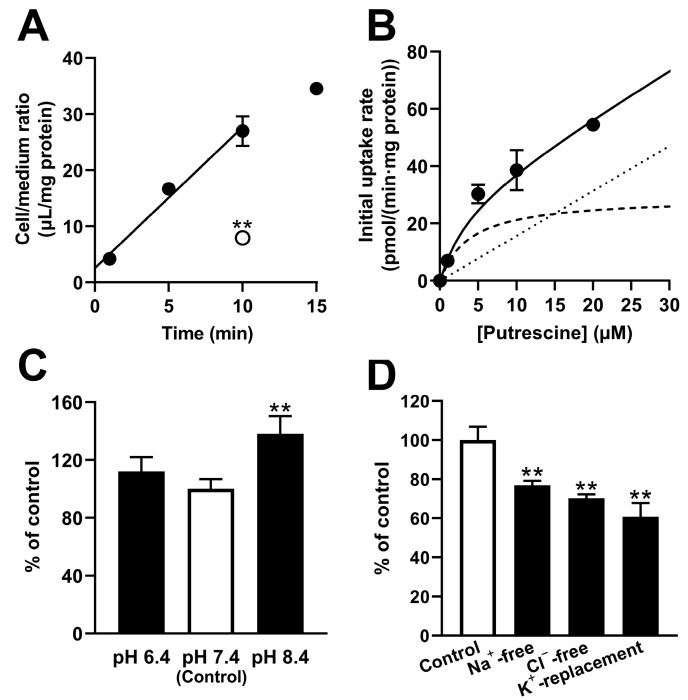
Uptake of [^3^H]putrescine by RPE-J cells. (**A**) The uptake of [^3^H]putrescine (0.1 μCi) was assessed at 37 °C (closed circles) and 4 °C (open circles). (**B**) The concentration-dependent uptake of [^3^H]putrescine (0.1 μCi) was examined at 37 °C for 10 min at concentrations in the range of 6.25 nM−20 μM. The data obtained in the uptake assay were plotted in the Michaelis–Menten analysis. The solid, dashed, and dotted lines represent the overall, saturable, and nonsaturable uptake, respectively. The dependence of [^3^H]putrescine uptake on extracellular pH (**C**), and on Na^+^, Cl^−^, and membrane potential (**D**), was examined at 37 °C for 10 min. The points and columns represent the means ± S.D. (*n* = 3). ** *p* < 0.01 vs. control.

**Figure 4 ijms-24-09003-f004:**
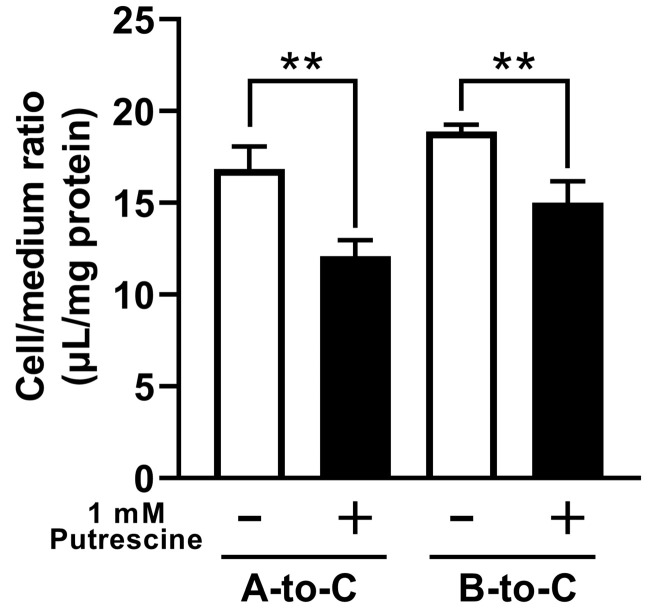
Directional uptake of [^3^H]putrescine by RPE-J cells on Transwell^®^ inserts. Uptake values of [^3^H]putrescine in apical-to-cell (A-to-C, 0.5 μCi) and basal-to-cell (B-to-C, 1.5 μCi) directions was assessed with or without unlabeled putrescine (1 mM) at 37 °C for 30 min. The columns represent the means ± S.D. (*n* = 3). ** *p* < 0.01 vs. the uptake in the absence of 1 mM putrescine.

**Figure 5 ijms-24-09003-f005:**
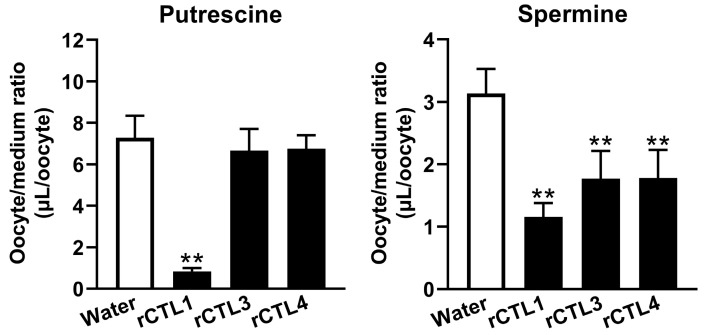
Uptake of polyamine by *Xenopus laevis* oocytes. The uptake of [^3^H]putrescine (0.45 μCi) and [^3^H]spermine (0.45 μCi) by oocytes injected with water, rCTL1, rCTL3, or rCTL4 cRNA was examined at 20 °C for 60 min. The columns represent the means ± S.D. (*n* = 10–15). ** * p* < 0.01 vs. water-injected oocytes.

**Figure 6 ijms-24-09003-f006:**
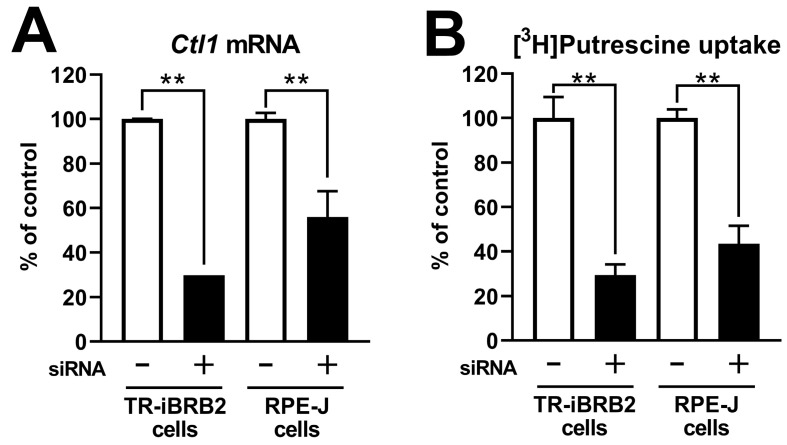
Knockdown analysis of [^3^H]putrescine uptake. TR-iBRB2 cells and RPE-J cells were transfected with negative control siRNA (−) or rCTL1 siRNA (+). (**A**) Quantitative real-time polymerase chain reaction analysis of rCTL1 took place in TR-iBRB2 cells and RPE-J cells, and the mRNA expression level of rCTL1 was normalized to that of β-actin. (**B**) The uptake of [^3^H]putrescine (0.1 μCi) by siRNA-transfected TR-iBRB2 cells and RPE-J cells was measured at 37 °C for 5 min and 10 min, respectively. The columns represent the means ± S.D. (*n* = 3). * *** *p* < 0.01 vs. the negative control siRNA-transfected cells.

**Table 1 ijms-24-09003-t001:** Elimination rate constant difference (Δβ) between [^3^H]putrescine and [^14^C]D-mannitol.

Compounds	% of Control
Control	100 ± 48
Putrescine	29.5 ± 11.7 *
Spermine	32.4 ± 30.5 *
TEA	76.7 ± 31.3

Each inhibitor (50 mM) was perfused in the microdialysis probe. The elimination of [^3^H]putrescine and [^14^C]D-mannitol was evaluated in the presence of putrescine, spermine, and TEA. The values represent the means ± S.D. (*n* = 3–5). * *p* < 0.05 vs. control. TEA—tetraethylammonium.

**Table 2 ijms-24-09003-t002:** In vivo uptake of [^3^H]putrescine by the retina in rats.

Compound	RUI (%)
Control	41.6 ± 3.0
Putrescine	33.5 ± 4.2 *

Rats were given a bolus injection of [^3^H]putrescine (5.0 μCi/rat) and [^14^C]*n*-butanol (0.5 μCi/rat), without (control) or with 50 mM putrescine, into the common carotid artery. The values represent the means ± S.D. (*n* = 3–4). * *p* < 0.05 vs. control. RUI—retinal uptake index.

**Table 3 ijms-24-09003-t003:** Inhibitory effects of several compounds on the uptake of [^3^H]putrescine.

Compound	Conc. (mM)	Relative Uptake (% of Control)	
		TR-iBRB2 Cells	RPE-J Cells
Control		100 ± 6	100 ± 6
Putrescine	1	10.1 ± 0.4 **	66.2 ± 3.6 **
Spermidine	1	10.4 ± 1.5 **	53.0 ± 3.9 **
Spermine	1	11.4 ± 1.3 **	50.9 ± 0.6 **
Agmatine	1	11.6 ± 0.7 **	55.2 ± 1.2 **
Serotonin	1	22.4 ± 3.4 **	33.5 ± 2.3 **
TEA	1	35.1 ± 4.3 **	75.9 ± 3.4 **
TEA	20	42.4 ± 0.5 **	N.D.
L-Carnitine	1	43.8 ± 2.3 **	83.6 ± 3.7 **
Choline	5	57.7 ± 5.1 **	49.3 ± 3.5 **
L-Ornithine	1	62.2 ± 9.0 **	91.8 ± 7.2
MPP^+^	1	62.8 ± 1.3 **	65.6 ± 4.5 **
L-Arginine	1	65.4 ± 5.4 **	98.5 ± 5.3
Cimetidine	1	90.1 ± 1.8 *	88.7 ± 9.3 *
Pyrimethamine	0.01	N.D.	44.9 ± 2.8 **
Lopinavir	0.01	N.D.	76.5 ± 1.0 **
Fluvoxamine	0.01	N.D.	76.2 ± 3.3 **

The uptake of [^3^H]putrescine (0.1 μCi) was measured in the absence (control) or presence of the compounds at 37 °C. Uptake by TR-iBRB2 cells and RPE-J cells was examined for 5 and 10 min, respectively. The values represent the means ± S.D. (*n* = 3–9). * *p* < 0.05 and ** *p* < 0.01 vs. control. MPP^+^—1-methyl-4-phenylpyridinium; N.D.—not determined.

## Data Availability

Upon reasonable request, data from this study are available from the corresponding author.

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
