# Peer review of "Carrier-Mediated Process of Putrescine Elimination at the Rat Blood–Retinal Barrier"

_ijms, 2023, doi:10.3390/ijms24109003_

Round 1

Reviewer 1 Report

Paper about a putrescine, an important molecule for many cell types including retinal cells. The level of this molecule must be controlled exactly because both low as well as to high levels are  deleterious for the function. The submitted manuscript is quite sane in nearly all aspects.

However, in discussion the paper should also mention the paper of Derek Bowie in JBC 2018 because it deals with channel block  of the important ionotropic glutamate receptors by polyamides, as putrescine and those receptors are very important for excitatory neurotransmission.

Minor thing: please replace in line 59 seldom by rarely

English language is fine - only the mentioned error should be corrected.

Reviewer 2 Report

Thanks for the opportunity to review the manuscript titled " Carrier-mediated process in putrescine elimination at the rat blood–retinal barrier”. This manuscript is a nice and interesting paper. The research topic is actual and useful for clinical and biomedical applications. The introduction, results and conclusion are well written, and all it should be of great interest to the readers. 

In my view this is ready for publication but may need some minor changes.

1.      There is no statistical analysis section, and the authors report significant differences in the results.

2.     In section 4.1 of the committee's ethical approval, there is no specific code for this procedure with animals.
